# Dolichos Lablab Linné Inhibits Bone Density Loss and Promotes Bone Union in Senile Osteoporosis through Osteogenesis

**DOI:** 10.3390/ph16101350

**Published:** 2023-09-25

**Authors:** Minsun Kim, Jae-Hyun Kim, Sooyeon Hong, Sumin Lee, Seung Hoon Lee, Jun Won Choi, Hyuk-Sang Jung, Youngjoo Sohn

**Affiliations:** Department of Anatomy, College of Korean Medicine, KyungHee University, Seoul 02-447, Republic of Korea; alstjs8644@naver.com (M.K.); jhk1@khu.ac.kr (J.-H.K.); ghdtndus121@naver.com (S.H.); 00sumin0209@naver.com (S.L.); sknight2230@naver.com (S.H.L.); wnsdnjs5805@gmail.com (J.W.C.)

**Keywords:** Dolichos lablab Linné, osteoblast, senile osteoporosis, fractured model, BMP-2, Wnt

## Abstract

As populations continue to age, osteoporosis has emerged as an increasingly critical concern. Most advancements in osteoporosis treatment are predominantly directed toward addressing abnormal osteoclast activity associated with menopause, with limited progress in developing therapies that enhance osteoblast activity, particularly in the context of aging and fractures, and serious side effects associated with existing treatments have highlighted the necessity for natural-product-based treatments targeting senile osteoporosis and fractures. Dolichos lablab Linné (DL) is a natural product traditionally used for gastrointestinal disorders, and its potential role in addressing bone diseases has not been extensively studied. In this research, we investigated the anti-osteoporosis and bone-union-stimulating effects of DL using the SAMP6 model, a naturally aged mouse model. Additionally, we employed MC3T3-E1 cells to validate DL’s osteoblast-promoting effect and to assess the involvement of core mechanisms such as the BMP-2/Smad and Wnt/β-catenin pathways. The experimental results revealed that DL promoted the formation of osteoblasts and calcified nodules by upregulating both the BMP-2/Smad and Wnt/β-catenin mechanisms. Based on its observed effects, DL demonstrated the potential to enhance bone mineral density in aged osteoporotic mice and promote bone union in fractured mice. These findings indicate the promising therapeutic potential of DL for the treatment of osteoporosis and bone-related conditions, thus warranting further investigation and potential clinical applications.

## 1. Introduction

Osteoporosis, a significant health risk characterized by low bone mass and increased fracture susceptibility, burdens society with the occurrence of osteoporotic fractures [1]. As populations age, osteoporosis becomes an increasingly critical concern. This condition develops due to the imbalance between bone-resorbing osteoclasts and bone-forming osteoblasts, disrupting normal bone metabolism [2]. Understanding the roles of osteoclasts and osteoblasts and their dysregulation is crucial for comprehending the pathogenesis of osteoporosis and addressing the challenges associated with its rising prevalence. Most advancements in osteoporosis treatment are predominantly directed toward addressing abnormal osteoclast activity associated with menopause, with limited progress in developing therapies that enhance osteoblast activity, particularly in the context of aging and fractures [3]. This narrow concentration overlooks other crucial segments, such as male patients, premenopausal women, and individuals with secondary osteoporosis caused by underlying medical conditions or medication usage [4]. Contemporary osteoporosis treatments predominantly target the attenuation of osteoporosis progression through the administration of bisphosphonate agents and selective estrogen receptor modulators (SERMs) [5]. However, these treatment options primarily target the suppression of osteoclast activity, and their efficacy in treating osteoporosis associated with conditions other than postmenopausal hormone deficiency is limited [6]. Moreover, these medications are often associated with significant side effects, including gastrointestinal disturbances, musculoskeletal pain, and rare but serious complications such as atypical femoral fractures and osteonecrosis of the jaw [7]. Consequently, this restricted approach fails to address the diverse and evolving needs of the osteoporosis patient population.

The development of therapeutic interventions for age-related osteoporosis is of great significance, given the prevalence and impact of this condition on the elderly population [1]. Effective treatments that can halt or reverse bone loss and prevent fractures are crucial for improving the quality of life and reducing the healthcare burden associated with osteoporosis. To achieve this, it is essential to establish reliable and relevant animal models that accurately replicate the characteristics of human osteoporosis. Senescence-accelerated mouse strain P6 (SAMP6) mice present a compelling choice for studying age-related osteoporosis and developing potential treatments [8]. The progressive decline in bone strength in SAMP6 mice closely resembles the clinical manifestation of osteoporosis in humans. This similarity makes SAMP6 an ideal model for investigating the underlying mechanisms of osteoporosis and evaluating the efficacy of potential therapeutic interventions [8].

Dolichos lablab Linné (DL) has predominantly been utilized as a countermeasure against poisoning and as a therapeutic intervention for diverse gastrointestinal disorders, encompassing cholera, diarrhea, and abdominal discomfort [9]. Recent investigations have demonstrated the efficacy of DL in ameliorating obesity [10] and alcoholic fatty liver disease [11]. However, the impact of DL on osteoblast activation and its potential in treating bone ailments remains unexplored. This research aims to validate and elucidate the efficacy of DL in preventing the loss of bone mineral density and fracture healing by activating the bone morphogenetic protein 2 (BMP-2)/SMAD family member 1 (Smad) [12] and Wnt/β-catenin pathways [13], which serve as key mechanisms for osteoblast differentiation.

## 2. Results

### 2.1. DL Promotes Osteoblastogenesis and Calcified Nodule Formation through Activation of BMP-2/Smad and Wnt/β-Catenin Mechanisms

To assess the impact of drugs on MC3T3-E1 cytotoxicity, cellular viability was evaluated using the cell counting kit-8 (CCK-8) assay following treatment with DL for 1 and 3 days. As depicted in Figure 1A,B, DL exhibited no toxicity within the concentration range of 25 to 100 µg/mL on both the 1st and 3rd days. When treated with 200 µg/mL of DL for one day, certain results indicated minimal cytotoxicity; therefore, for the experiments in this study, 100 µg/mL was chosen as the maximum concentration. These findings indicate that drug reactions observed in cell experiments are unrelated to cytotoxic effects. To corroborate the differentiation and functionality of DL in osteoblasts, MC3T3-E1 cells were treated with osteogenic medium and DL for a duration of 14 days, and subsequently, the calcified nodules formed on the cells were stained with Alizarin red S and Von Kossa. Staining analysis confirmed a concentration-dependent increase in DL, while only minimal calcified nodules were observed in the control group (cells treated with osteogenic medium alone) (Figure 1C). Moreover, quantification of Alizarin red S dye extraction and absorbance measurement revealed a promotion of nodule formation in a concentration-dependent manner (Figure 1D). The efficacy of DL in promoting the expression of the BMP-2/Smad signaling pathway and runt-related transcription factor 2 (RUNX2), a pivotal transcription factor in osteoblasts [12], was validated through Western blot analysis. The expression levels of BMP-2, Osterix, phosphorylated Smad1/5 (P-Smad1/5), and RUNX2 exhibited a tendency to increase following DL treatment compared to the control group (Figure 1E). Upon quantification of each marker’s expression relative to Actin (P-Smad1/5 was quantified using Smad1/5/9), significant increases were observed in BMP-2 (from 50 µg/mL), Osterix (from 12.5 µg/mL), p-Smad (100 µg/mL), and RUNX2 (from 25 µg/mL), in comparison to the control (Figure 1F). Furthermore, the impact of DL on mRNA expression related to the Wnt/β-catenin signaling pathway was assessed using reverse transcription-polymerase chain reaction (RT-PCR). The expression levels of *Ctnnb1* (β-catenin), *Wnt10b*, Segment polarity protein disheveled homolog DVL-2 (*Dvl2)*, and Cyclin D1 (*Ccnd1)* exhibited an increasing trend with DL treatment compared to the control group (Figure 1G). Quantification of each marker’s expression normalized to *Gapdhs* revealed significant increases in *Ctnnb1* at 100 µg/mL, *Wnt10b* at 25 and 100 µg/mL, *Dvl2* at 50 and 100 µg/mL, and *CCnd* at 100 µg/mL, when compared to the control (Figure 1H).

### 2.2. DL Induces the Activation of BMP-2/Smad and Wnt/β-Catenin Pathways

The upregulation of these two mechanisms regulates the expression of alkaline phosphatase (ALP, *Alpl*), osteocalcin (OCN, *Bglap2*), bone sialoprotein (BSP, *Ibsp*), and pro-alpha1 chains of type I collagen (procol1, *Col1a1*), which are involved in the early and intermediate stages of osteoblast differentiation and contribute to the process of calcification, nodule formation, and bone matrix formation [14]. Consistent with the previous findings, DL treatment resulted in an increased mRNA expression of *Alpl*, *Bglap2*, *Ibsp*, and *Col1a1* compared to the control group (Figure 2A). Quantifying the expression of each marker normalized to Gapdhs, *Alpl* showed significant effects starting from 25 μg/mL, *Bglap* showed significant effects starting from 50 μg/mL, and *Ibsp* and *Col1a1* showed significant effects at all concentrations (Figure 2B).

### 2.3. DL Positively Regulates Changes in Osteogenesis Markers in a Senile Osteoporosis Mouse

SAMP6 mice are characterized by a decrease in bone density and an increase in body weight as mesenchymal cells are differentiated into adipocytes rather than osteoblasts due to natural aging [8]. As shown in Figure 3A, the weight of the SAMP6 group increased significantly compared to the senescence-accelerated mouse-resistant 1 (SAMR1) group. Administration of DL-H had an inhibitory effect on body weight gain. In addition, the effects of aging and DL were confirmed on liver weight after sacrifice and AST and ALT, which are hepatotoxicity indicators in serum (Figure 3B–D). In all three data, DL did not show a significant effect compared to the SAMP6 group. These results are thought to have been derived from similar results in animal experiments on DL, which was non-toxic in in vitro experiments. The impact of DL on the expression of serum markers related to bone formation was confirmed. In the SAMP6 group, there was a reduction in the expression levels of ALP and OCN compared to the SAMR1 group, while the DL high-dose administration group effectively attenuated these reductions. Leptin levels were found to be elevated in SAMP6. However, administration of DL-H significantly attenuated this increase. The expression of Sclerostin (SOST) was markedly reduced in SAMP6, and DL-H treatment successfully reversed this reduction. Furthermore, the expression of Dickkopf WNT signaling pathway inhibitor (DKK-1), an antagonist of the Wnt signaling pathway, was elevated in the SAMP6 group, but DL-H exhibited a strong inhibitory effect on this increase (Figure 3E–I).

### 2.4. DL Suppresses the Decrease in Bone Density in the Femur of a Senile Osteoporosis Mouse

The microcomputed tomography (micro-CT) analysis method enables the assessment of changes in overall bone quality by examining images and data that go beyond simple bone density, including the analysis of intra-bone microstructure [15]. The bone density of the femur, measured after sacrifice, was found to be reduced in the SAMP6 group compared to the SAMR1 group. However, administration of DL at high concentrations resulted in bone density similar to that of the SAMR1 group (Figure 4A). Histological analysis further supported these findings, demonstrating that DL effectively suppressed the reduction in cancellous bone area observed in the SAMP6 model (Figure 4B). Micro-CT analysis of the bone volume/total volume (BV/TV) revealed an increase in the SAMP6 group following DL administration, consistent with the visual observations (Figure 4C). Trabecular thickness (Tb.Th), trabecular separation (Tb.Sp), and trabecular number (Tb.N) are microstructural parameters used to assess cancellous bone quality. These indicators play a crucial role in determining bone health. In this study, SAMP6 mice exhibited noteworthy distinctions in each of these indicators when compared to SAMR1 mice. Additionally, mice treated with DL displayed favorable alterations in all three indicators compared to the induced group (Figure 4D–F). The analysis of cancellous bone area also showed a consistent trend with the aforementioned findings (Figure 4G).

### 2.5. DL Shows Positive Changes in Bone Morphogenetic Factors in Serum in Geriatric Fracture Animal Models and Promotes Bone Union by Inducing the Formation of Osseous Callus

To induce senile osteoporosis, a similar protocol was employed as in the previous experiment. Thereafter, a fracture was induced in the femur using an electric saw. Subsequently, DL was administered, and the mice were sacrificed at 2 and 4 weeks post-administration to analyze the effectiveness in both the femur and serum. DL did not show a significant change in the expression of AST/ALT in serum (Figure 5A,B). Notably, DL administration resulted in a significant increase in the expression of ALP in serum at 2 weeks and the expression of OCN at 4 weeks, respectively (Figure 5C,D). To assess the bone union efficacy of DL, a comparison was made between the bone-union-promoting effect of the induced group and the DL treatment group at 2 weeks and 4 weeks, respectively (Figure 5E). The administration of DL for 4 weeks led to a significant increase in fracture line and callus formation at the fractured femur. Additionally, when analyzing the microstructure of the fracture site using micro-CT software, significant effects were observed at 4 weeks for BV (bone volume) and at 2 and 4 weeks for BMD (bone mineral density) and Tb.Tb. However, Tb.Sp showed a positive change at 4 weeks, but the difference was not statistically significant (Figure 5F–I). The rapid and substantial formation of the osseous callus plays a crucial role in evaluating the efficacy of bone union. While no significant change was observed in the H&E staining data after 2 weeks of DL treatment, mice administered with DL for 4 weeks exhibited accelerated osseous callus formation compared to SAMP6 mice (Figure 5J).

### 2.6. Standardization of DL by Liquid Chromatography-Mass Spectrometry (LC-MS) Analysis

In the context of establishing drug efficacy reproducibility and stability, it is crucial to provide data pertaining to the quality control and quantification of natural product samples [16]. With this in mind, the prominent constituents of DL were subjected to peak analysis using L-arginine and Trigonelline as reference compounds [17]. The experimental findings unequivocally confirmed the simultaneous detection of peaks corresponding to L-arginine and Trigonelline in DL (Figure 6A,B).

## 3. Discussion

The experimental findings of this study delineated the distinct pharmacological functions of DL in the context of senile bone disease. Firstly, DL exhibits a noteworthy capacity in enhancing osteoblast differentiation and facilitating the formation of calcified nodules. Secondly, these effects are orchestrated through the intricate involvement of the BMP-2/Smad and Wnt/β-catenin mechanisms. Thirdly, DL effectively impedes the age-related decline in bone density within the femur. Lastly, DL significantly enhances the process of bone union, particularly during the early and intermediate stages of femoral fracture.

MC3T3-E1 cells are a lineage of osteoblastic cells derived from the skull of C57BL/6 mice [18]. Within this cell line, there exist three distinct subclones (4, 14, and 24), with subclone 4 exhibiting a fibroblast-like morphology while displaying elevated ALP activity following long-term culture. These cells possess the capability to differentiate into both osteoblasts and osteocytes, contributing to the process of bone matrix calcification [19].

Mesenchymal stem cells (MSCs) possess the capacity to differentiate into osteoblasts, adipocytes, myoblasts, and chondroblasts. Throughout the differentiation process, the activation of transcription factors such as BMP-2 and Wnt triggers the expression of RUNX2, ALP, OCN, BSP, and procol1, ultimately leading to the differentiation of MSCs into osteoblasts [20]. However, as these indicators are not exclusively limited to osteoblasts, most pharmacological effect verification studies initially focus on assessing the effects of osteoblast differentiation and bone formation by examining the mineralization of the extracellular matrix [21]. In this study, we investigated the mineralization-promoting effect of DL using alizarin red S and Von Kossa staining. The results revealed enhanced mineralization across all concentrations of DL-treated cells. The transcription factor RUNX2 plays a key role in osteoblast differentiation and is regulated through two mechanisms [22]. Firstly, the BMP-2/Smad mechanism involves the activation of BMP-2, which induces Smad phosphorylation, leading to nuclear translocation [12]. This, in turn, activates RUNX2 and osterix. Secondly, the Wnt/β-catenin signaling is initiated by the secretion of Wnt ligands from the frizzled receptor domain of Lrp5/6 receptors [13]. This activation triggers a downstream signaling cascade that inhibits the cytoplasmic glycogen GSK-3β [13]. As a result, β-catenin, a central mediator of canonical Wnt signaling, is released from inhibition. The Wnt/β-catenin signaling pathway plays a crucial role in stimulating osteoblastogenesis by promoting RUNX2 expression and the differentiation of pluripotent BMSCs (bone-marrow-derived mesenchymal stem cells) into the osteoblast lineage [23]. Simultaneously, it inhibits commitment to the cartilage and adipogenic lineages, thereby directing cell fate toward osteoblast formation. Subsequently, RUNX2 regulates the expression of crucial factors involved in osteoblast differentiation and activity, such as ALP, OCN, and BSP [22]. The significance of RUNX2 has been extensively demonstrated in various studies. Inada’s study showed that mice lacking the Runx2 gene (Runx2 –/–) exhibit a lack of osteoblasts and impaired bone formation, with notable inhibition of chondrocyte maturation [24]. Additionally, Komori et al. demonstrated that Runx2 –/– mice lack osteoblasts, exhibit no expression of bone matrix protein genes (including *Spp1, Ibsp*, and *Bglap2*), and display significantly reduced Col1a1 expression in the presumed bone region [25]. The findings of this study demonstrate that DL effectively upregulated both mechanisms and consequently led to a significant increase in the expression of RUNX2. These results strongly suggest a direct association between DL’s mineralization-promoting effect and the underlying mechanisms involved.

ALP, found in bone tissue, is primarily expressed during the early stages of osteoblast differentiation, and its production increases with bone growth [26]. It is present on the surface and stromal vesicles of osteoblast cells and acts as a regulator of inorganic phosphate transport, cell division, and differentiation during osteoblast mineralization. ALP also serves as a biomarker for evaluating cell phenotype and developmental maturity [26]. OCN, specifically expressed in osteoblasts, is the most abundant non-collagenous protein in bone. It plays a critical role in bone metabolism and serves as a clinical marker for bone turnover [27]. OCN is involved in regulating the crystal form and growth of hydroxyapatite, a key component of bone mineralization. BSP is a component of mineralized tissues, including bone, dentin, cementum, and calcified cartilage. It constitutes approximately 8% of all non-collagen proteins [28]. While the precise role of BSP in the mineralized structure is not fully understood, it is hypothesized to act as a nucleation site for the formation of apatite crystals. Col1a1, the most abundant protein in mammals, contributes to 90% of the total organic component of the bone matrix. However, similar to BSP, the specific cellular origin and functional contribution of Col1a1 in bone formation have not been fully elucidated [29]. Nonetheless, genetic mutations in either the Col1a1 or Col1a2 genes in humans can result in various subtypes of osteogenesis imperfecta syndrome, commonly known as brittle bone disease. In this study, DL treatment significantly induced the expression of ALP, OCN, BSP, and Col1a1 in osteoblasts. This effect is likely mediated by the synergistic action of upregulated RUNX2 expression in response to DL. These findings suggest that the observed promotion of calcification is a result of the upregulation of these factors. Collectively, the cell experiment findings indicate that DL may have potential applications in elderly osteoporosis patients and individuals with fractures requiring enhanced osteoblast activity. DL elicits the expression of calcific nodules and osteogenic factors at a relatively low concentration (12.5 µg/mL) and exhibits an upward trend across all concentrations tested. Notably, at 100 µg/mL, a significant impact was observed in all experiments when compared to cells treated with the osteogenic medium. However, it is noteworthy that the cell viability test results indicated an effect on cell viability at 200 µg/mL. This concern is believed to be addressable through subsequent experiments and future in vivo toxicity assessments.

In comparison to typical fractures, osteoporotic fractures can manifest even with minimal force and exhibit a protracted recovery period, resulting in a diminished quality of life and potentially fatal complications [30]. However, due to the absence of overt clinical signs in osteoporosis, a substantial number of patients fail to recognize the imperative for timely treatment, consequently placing them at an elevated risk for osteoporotic fractures [31]. Currently, the prevailing focus of research in general osteoporosis drug development primarily revolves around attenuating bone density through the inhibition of osteoclast activity or augmenting bone density via osteogenesis. Nevertheless, given the substantial portion of patients at risk of fracture or directly experiencing fractures, it is of paramount clinical significance to ascertain the anti-osteoporotic efficacy of the treatment and its potential to enhance bone union. Therefore, in this study, both the anti-osteoporotic effect and the bone-union-promoting effect of DL in geriatric osteoporotic animals were verified.

The process of bone aging can lead to the development of various degenerative skeletal conditions, such as primary osteoporosis and osteoarthritis. Additionally, it can adversely affect other systems in the body, including the central nervous system, hematopoietic and immune systems, and the endocrine system. In 1975, the accidental mating of AKR/J mice with an unidentified albino mouse strain led to the development of the precursor strain known as SAM. Among the SAM strains, the SAMP6 mouse was initially identified as an animal model for naturally occurring age-related osteoporosis in 1981 [8]. This strain exhibits specific characteristics resembling human senile osteoporosis, including a reduced rate of bone formation, decreased numbers of bone marrow progenitor cells, diminished bone strength, thin cortical bone, enlarged periosteum, and decreased inner bone diameter. Due to these similarities, the SAMP6 mouse has been extensively employed in various studies related to osteoporosis. A notable distinction between SAMP6 and SAMR1 mice is observed in the cancellous bone density of the femur starting from 3 months of age [32]. It has been hypothesized that the mechanism underlying osteoporosis induction in SAMP6 mice involves the Wnt signaling pathway, which subsequently suppresses osteoblast activity and bone formation [33]. The findings of this study demonstrate that DL significantly mitigated the reduction in femoral bone density in SAMP6 mice. Moreover, analysis of the bone microarchitecture index (Tb.Tb, Tb.Sp, and Tb.N) indicated positive alterations in the composition of bone microstructure, including bone density. Furthermore, DL effectively modulated the levels of osteoblast-related factors, namely ALP, OCN, and DKK-1, in the serum. These results align with those obtained from in vitro experiments, reinforcing the notion that DL exerts an anti-osteoporotic effect through the activation of osteoblasts.

Furthermore, we assessed the efficacy of DL in a femoral fracture model. Fracture healing involves a dynamic and intricate sequence of events, including inflammation, restoration, and remodeling [34]. Thus, determining the optimal treatment duration to evaluate the drug’s effectiveness is of paramount importance. In this study, we specifically aimed to investigate the osteoblast-activation-promoting effect of DL. Therefore, we designated the reconstructive-remodeling period, which spans 2 to 4 weeks after fracture and encompasses callus formation and enhanced bone formation, as the experimental endpoint. The experimental findings revealed significant outcomes regarding the effect of DL on bone union promotion. Micro-CT analysis, combined with image and microstructure assessment, demonstrated that DL expedited callus formation within 2 weeks and exhibited a bone-union-promoting effect at 4 weeks. This result was consistent with the histological analysis using H&E staining, which revealed rapid formation of fibrous callus followed by its transformation into osseous callus, demonstrating the efficacy of DL in facilitating bone union. Serum analysis further supported these findings, showing a significant increase in ALP expression at two weeks and OCN expression at four weeks. Considering the established knowledge that ALP serves as an early bone marker and OCN levels are elevated during the intermediate stages, it can be concluded that DL effectively promotes bone union through stimulating osteoblast activity.

## 4. Materials and Methods

### 4.1. Reagents

Minimum essential medium-α (α-MEM) and penicillin/streptomycin (P/S) were obtained from Gibco (Gaithersburg, MD, USA). Cell counting kit-8 (CCK-8) assay was obtained from Dojindo Molecular. Fetal bovine serum (FBS) was procured from Atlas Biologicals (Fort Collins, CO, USA). Ascorbic acid, β-glycerophosphate, dimethyl sulfoxide (DMSO), Na_2_S_2_O_3_, and AgNO_3_ were obtained from Sigma-Aldrich (St. Louis, MO, USA). BMP-2, RUNX2, SMAD1/5/9, and osterix were purchased from Abcam (Cambridge, UK). p-SMAD1/5 was sourced from Cell Signaling Technology, Inc. (Danvers, MA, USA). Beta-actin from santacruz (St. Louis, MO, USA) was purchased. PCR primers were obtained from genotech (Daejeon, Republic of Korea), Taq polymerase from Kapa Biosystems (Woburn, MA, USA), and RTase and SYBR green from Invitrogen (Carlsbad, CA, USA). The OCN kit (cat. no: LS-F12230) was acquired from LSbio (Seattle, WA, USA).

### 4.2. Ethanol DL Extract Preparation

DL was purchased from Omniherb (Seoul, Republic of Korea). A total of 200 g of DL was cooled in 2 L of 30% Ethanol for a duration of 3 weeks. Following this, the solvent was substituted with deionized water (DW) using a concentrator and subsequently subjected to freeze-drying, resulting in a yield of 6.4%. The obtained powder was stored at −20 °C until required and was dissolved in DMSO before application. In cell experiments, the amount of DMSO utilized did not exceed 0.1% of the total dosage.

### 4.3. Osteoblast Differentiation Model and Verification of Cytotoxicity of DL

MC3T3-E1 cells were procured from the American Type Culture Collection (ATCC). The growth medium used for cell culture consisted of α-MEM without ascorbic acid, supplemented with 10% FBS and 1% P/S in a cell incubator maintained at 37 °C and 5% CO_2_. Subculturing was carried out every 2 days. For osteoblast differentiation, an osteogenic medium was employed, which comprised α-MEM without ascorbic acid, supplemented with 10% FBS and 1% P/S, 25 µg/mL ascorbic acid, and 10 mM β-glycerophosphate. To assess the cytotoxicity of DL, we seeded 50,000 cells in a 96-well plate and allowed them to stabilize overnight. Subsequently, the cells were cultured for 1 day and 3 days in a-MEM medium (without FBS and ascorbic acid) supplemented with varying concentrations of DL (ranging from 25 to 400 µg/mL). After that, 10 µL of CCK-8 solution was added to each well, and the cells were incubated for 1 h in a cell incubator to facilitate the reaction. Cell viability was determined by converting the plate data to absorbance measurements at 490 nm using an enzyme-linked immunosorbent assay (ELISA) machine. The results were expressed as a percentage relative to untreated cells. Moreover, any condition resulting in cell viability below 90% was considered indicative of cytotoxicity.

### 4.4. Alizarin Red S Staining and Von Kossa Staining

To evaluate the impact of DL on the formation of mineralized nodules, we seeded 15,000 MC3T3-E1 cells into a 24-well plate. Subsequently, we replaced the medium with osteogenic medium and cultured the cells for 14 days. The medium was changed every 2 days using the same medium. Upon completion of the incubation period, the cells were washed three times with cold DPBS and then fixed with 80% Et-OH at 4 °C for 1 h. Next, the plate was stained with Alizarin red S solution at room temperature for 5 min, and we captured images of the stained plate using a digital camera. To quantify the stained areas, the dye was extracted using 10% (*v*/*w*) cetylpyridinium chloride (CPC; Sigma-Aldrich, St. Louis, MO, USA), and the absorbance was measured at a wavelength of 570 nm using an ELISA reader device. For the von Kossa staining, 1% AgNO_3_ was added to the fixed plate and exposed to UV light for 40 min. Following this, the plate was treated with 5% Na_2_S_2_O_3_, washed with distilled water (DW), and then dried. The von Kossa-stained area on the plate was quantified using ImageJ version 1.46 software, and images were taken using a digital camera.

### 4.5. Western Blot

To assess the influence of DL on the BMP-2/Smad signaling pathway, we seeded 50,000 MC3T3-E1 cells into a 60 mm culture dish for 2 days. The cells were subsequently lysed using radioimmunoprecipitation assay (RIPA) buffer containing 50 mM Tris-Cl, 150 mM NaCl, 1% NP-40, 0.5% Na-deoxycholate, and 0.1% SDS, supplemented with protease inhibitors and phosphatase inhibitors 2 and 3. Afterward, the lysates were centrifuged at 13,200 rpm for 20 min to obtain the supernatant containing total protein. The protein concentrations were determined using BCA solution. Equal amounts of protein (30 μg) were separated by electrophoresis on a 10% sodium dodecyl sulfate-polyacrylamide gel electrophoresis (SDS-PAGE) and then transferred to a nitrocellulose transfer membrane (Whatman Protran, Dassel, Germany). Non-specific binding sites on the membrane were blocked using 5% skim milk (blocking buffer) for 1 h. Primary antibodies, diluted 1:1000–1:2000 in 1% bovine serum albumin (BSA), were applied to the membrane and incubated at 4 °C for 24 h. The detailed information and dilution factor of the antibody are listed in Table 1. Subsequently, the membrane was exposed to a secondary antibody (peroxidase-conjugated anti-IgG), diluted 1:10,000 in blocking buffer, and incubated at room temperature for 1 h. Afterward, the membrane was washed 6 times for 10 min each with Tris-buffered saline with tween 20 (TBST) solution. Membranes were expressed on X-ray film using an enhanced chemiluminescence (ECL) kit (Santa Cruz, CA, USA) solution and imaged using Image J software (Ver 1.46, National Institute of Health, Washington, DC, USA). Bands were quantified.

### 4.6. RT-PCR

To validate the impact of DL on the Wnt/β-catenin signaling pathway and the expression of osteoblast-related genes, we employed RT-PCR analysis. Firstly, we seeded 30,000 MC3T3-E1 cells into a 6-well plate and cultured them for 4 days. The culture medium was replaced with fresh medium every 2 days during the incubation period. Upon completion of the incubation, the cells were washed three times with DPBS, and total RNA was extracted using TRIZol. Subsequently, the RNA was quantified, and 2 μg of RNA was used for cDNA synthesis employing the Superscript II reverse transcriptase kit. The synthesized cDNA was then subjected to amplification using Taq polymerase and specific primers listed in Table 2. The amplified cDNA products were separated on a 1.2% agarose gel stained with SYBR green, and the resulting bands were quantified using Image J software. This analysis allowed us to assess the influence of DL on the expression of osteoblast-related genes and the Wnt/β-catenin mechanism at the molecular level.

### 4.7. Establishment of Senile Osteoporosis Model and Verification of Anti-Osteoporosis Effect of DL

The experimental animals used in this study were 4-week-old males, SAMR1 and SAMP6, obtained from Central Lab. Animal Inc. (Seoul, Repibic of Korea), with an average weight of 25 g. After arrival, the rats were allowed to acclimate to the laboratory environment for 1 week. During the experiment, the animals were housed in an animal breeding room with a controlled temperature of 22 ± 3 °C, humidity maintained at 55–60%, and a 12 h light–dark cycle. Free access to feed and water was provided to all animals. The animal experiments were conducted following the guidelines of the Animal Experimentation Ethics Committee of Kyung Hee University (KHSASP-20-125). To induce a geriatric osteoporosis model, the rats were bred for 23 weeks. During this period, the body weight of the experimental animals was measured weekly at the same time. There were 8 animals in each group, and the drug administration was performed once daily at the same time throughout the experiment. The dosage of the DL administered was determined based on the principles of oriental medicine. The daily dosage for an adult (considering 60 kg) is 8 g. Thus, when 8 g was substituted with DL (yield: 6.4%), the administered dose was 0.384 mg/kg of DL. Considering that the metabolism of rats is approximately 12.3 times higher than that of humans, the DL-H group received 4.72 mg/kg of DL. During the breeding period, specific humane endpoints were established for the well-being of the animals, including (i) uncomfortable walking and difficulty accessing food or water; (ii) difficulty maintaining normal posture due to weakness; (iii) >20% weight loss compared to age-matched controls; (iv) symptoms of respiratory distress, cyanosis, chronic discomfort, or constipation; and (v) unconsciousness and unresponsiveness to external stimuli. However, all animals in the experiment did not experience any significant problems. At the end of the experiment, all animals were deeply anesthetized with 5% isoflurane and 100% O_2_, and blood was collected by lethal cardiac puncture. After confirming that the heart had stopped beating, the animals were euthanized by cervical dislocation. The femur and liver were then collected and weighed for further analysis.

### 4.8. Establishment of Senile Femoral Fractured Model and Verification of Bone-Union-Promoting Effect of DL

To explore the impact of bone union promotion in a senile fracture model, 4-week-old SAMP6 mice were utilized, which were obtained from Central Lab, Animal Inc. The animal experiments were conducted following the guidelines of the Animal Experimentation Ethics Committee of Kyung Hee University (KHSASP-21-387). The mice were reared under the same environmental conditions as the previous experiment, and they were bred for 20 weeks to induce senile osteoporosis. To cause a femur fracture, all animals were deeply anesthetized with 5% isoflurane and 100% O_2_. The hair at the surgical site was removed using a razor, and then the skin and quadriceps were incised with scissors to access the femur. The central part of the femur was subsequently cut using an electric saw, and it was stabilized with a K-Wire. The wound was closed by suturing the muscle and skin using thread. To prevent infection at the surgical site, intraperitoneal administration of gentamicin (4 mg/kg) was carried out for 3 days. Each group consisted of 8 animals. DL (4.72 mg/kg) was administered to the experimental groups for 2 and 4 weeks after the surgery. The control group (SAMP6) received an equal amount of distilled water. Throughout the experimental period, none of the mice reached the predefined humane endpoint. At the end of the experiment, all animals were deeply anesthetized with 5% isoflurane and 100% O_2_, and blood was collected by lethal cardiac puncture. After confirming that the heart had stopped beating, the animals were euthanized by cervical dislocation. The femur and liver were then collected and weighed for further analysis.

### 4.9. Serum Analysis

The collected blood was allowed to react at room temperature for 30 min and subsequently centrifuged at 2000 rpm and 4 °C for 10 min. The serum samples were then sent to DKkorea (Seoul, Republic of Korea), a serum analysis company, for measuring the content of AST, ALT, and ALP. Additionally, OCN was analyzed using an ELISA kit (LS-F12227, LSbio, WA, USA). Leptin, SOST, and DKK-1 were analyzed using the multi-assay kit from GClab (Seoul, Republic of Korea), another serum analysis company.

### 4.10. Micro-CT Analysis

Equipment Configuration: For femur scanning, a high-resolution cone-beam micro CT system (SkyScan1176, Skyscan, Kontich, Belgium) was utilized. The X-ray generator operated at 50 kV/200 μA, and the pixel size was set at 8.9 μm. Additionally, an aluminum (Al) filter of 0.5 mm was employed, and the scanning involved a rotation angle of 180° with rotation steps of 0.4°.

Assessment of Anti-Osteoporosis Effect: To investigate the anti-osteoporosis effect, femur heads were scanned in the proximal direction, commencing from the growth plate. The acquired images were captured and visualized using Data Viewer software (Skyscan v.1.5.1.9). Structural parameters were then extracted from the images using CT Analyzer software (Skyscan v1.15.4.0). The analysis focused on femoral neck trabecular lesions, extending 0.8 mm proximally from the growth plate’s end. Trabecular bone measurements were set with an offset of 0.1 mm and a height of 0.8 mm.

Assessment of Bone Union Promotion Effect: To evaluate the bone union promotion effect, fractured femurs were visualized through DataViewer software (Skyscan v.1.5.1.9). The microstructure of the callus was measured at the center of the cut site on 400 slides using Skyscan software (Skyscan v1.15.4.0). Subsequently, various bone morphometric parameters were quantified from the resultant three-dimensional images, including bone mineral density (BMD, g/cm^3^), bone volume fraction (BV/TV, %), trabecular thickness (Tb.Th, mm), trabecular separation (Tb.Sp, mm), and trabecular number (Tb.N, 1/mm).

### 4.11. Histomorphometric Analysis

Osteoporotic femurs and fractured femurs were fixed in neutral buffered formalin (NBF) and subsequently washed in running water for 24 h. Following this, they underwent decalcification with ethylenediaminetetracetic acid disodium (EDTA) for a period of 3 weeks. The samples were then degreased and dehydrated using ethanol and finally embedded in paraffin. Tissue sections of 5 μm thickness were obtained using a microtome (Carl Zeiss AG). The sections were then stained with hematoxylin and eosin (H&E), and the stained tissues were observed under a 100-fold magnification using an optical microscope. For analysis, the bone trabecular area was used as the standard, which included the area below the growth plate in the proximal femur, with the horizontal length being 2/3 of the growth plate length. In the case of the fractured femur, the fracture line was aligned at the center of the photograph. The bone density within the femur was measured using the image analysis program Image J, Ver. 1.46 (National Institutes of Health).

### 4.12. Liquid Chromatography-Mass Spectrometry (LC-MS) Analysis

DL extracts were subjected to analysis using LC-MS. A Prodigy™ ODS column (150 × 4.6 mm, 5 μm) was employed for separation. The flow rate was set at 0.3 mL/min, and the column temperature was maintained at ambient conditions throughout the 60 min run. The mobile phase consisted of acetonitrile and water with 0.1% formic acid. The elution gradient followed a programmed sequence of 5% B at 0 min, 5% to 50% B over 30 min, and 50% to 95% B over 60 min. Mass spectrometric data were acquired in positive scan mode, covering the mass range *m*/*z* 100–300 Da.

### 4.13. Statistical Analysis

The results were presented as mean ± standard error (mean ± S.E.M.), and statistical analysis was performed using Graph Pad Prism software (Ver. 5.01, Graph Pad Software Inc., La Jolla, CA, USA). Cell experiments were conducted a minimum of three times. To assess the differences between each group, a one-way ANOVA test was employed for comparison, and a significance level of *p* < 0.05 or less was considered statistically significant. Post hoc verification was carried out using Dunnett’s multiple comparison test.

## Figures and Tables

**Figure 1 pharmaceuticals-16-01350-f001:**
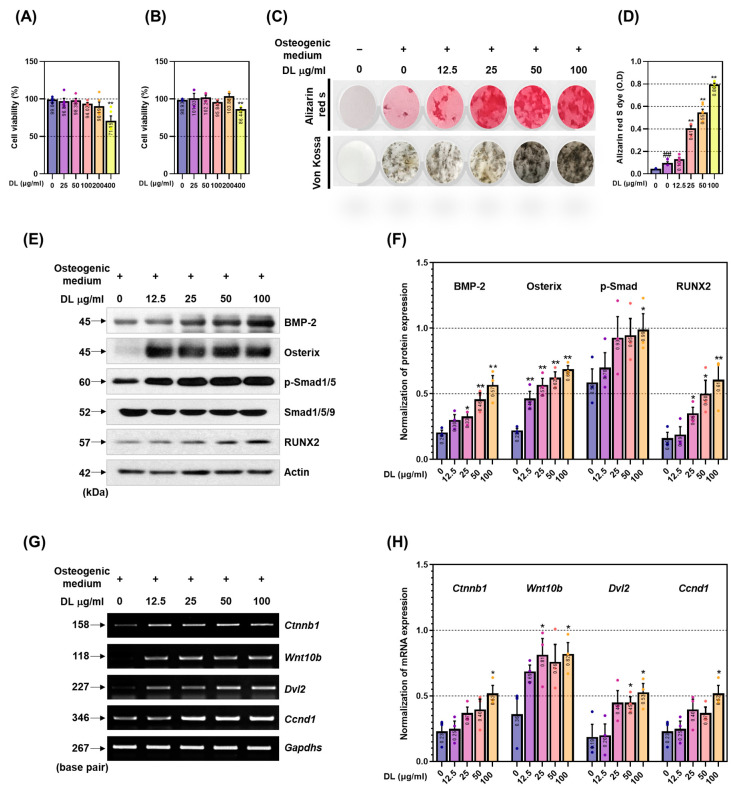
DL upregulates the BMP-2/Wnt mechanism, promoting osteoblast differentiation and calcified nodule formation. (**A**) Cell viability was measured using the CCK-8 kit 1 day after DL treatment and (**B**) 3 days after DL treatment. (**C**) The osteoblast-differentiation-promoting effect was verified through Alizarin red S staining. (**D**) The degree of staining was quantitatively analyzed at an absorbance of 450 nm after dye extraction. (**E**) DL’s effect on the BMP-2/Smad mechanism was verified using Western blot. (**F**) The expression of each factor was quantified by Actin or Smad1/5/9. (**G**) DL’s effect on the Wnt/B-catenin mechanism was verified using RT-PCR. (**H**) The expression of each factor was quantified by Gapdhs. All experiments were conducted at least three times. Dots in each graph represent individual experimental data values. Data are expressed as mean ± SEM. ## *p* < 0.01 indicates significance for cells not treated with anything; ** *p* < 0.01 and * *p* < 0.05 indicate significance for cells treated with osteogenic medium.

**Figure 2 pharmaceuticals-16-01350-f002:**
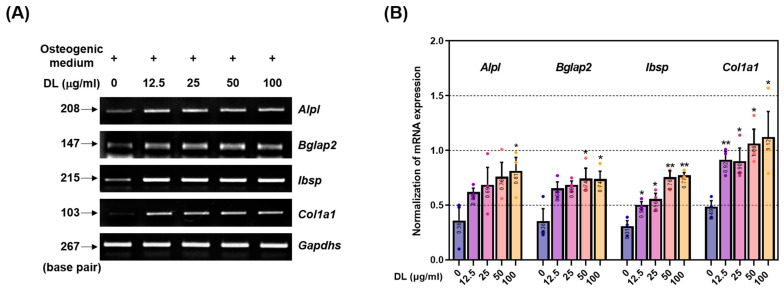
DL upregulates the osteoblast-related gene expression. (**A**) DL’s effect on the osteogenesis genes was verified using RT-PCR. (**B**) The expression of each factor was quantified by Gapdhs. All experiments were conducted at least three times. Dots in each graph represent individual experimental data values. Data are expressed as mean ± SEM. ** *p* < 0.01 and * *p* < 0.05 indicate significance for cells treated with osteogenic medium.

**Figure 3 pharmaceuticals-16-01350-f003:**
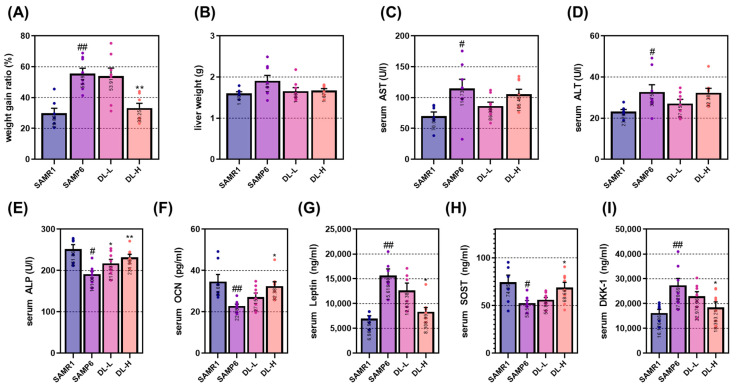
DL does not show toxicity due to long-term administration in mice with senile osteoporosis and causes positive changes in bone-related factors in serum. (**A**) The initial and final body weights were measured, and the rate of weight increase was compared. (**B**) Liver weights were measured after sacrifice to assess drug toxicity. (**C**,**D**) Serum levels of AST/ALT, known as indicators of hepatotoxicity, were measured using ELISA. ELISA was also employed to measure serum levels of bone-related factors such as (**E**) ALP, (**F**) OCN, (**G**) Leptin, (**H**) SOST, and (**I**) DKK-1. Each mouse group consisted of 8 mice. Dots in each graph represent individual experimental data values. Data are presented as mean ± SEM. ## *p* < 0.01 and # *p* < 0.05 indicate significance for the SAMR1 mice group; ** *p* < 0.01 and * *p* < 0.05 indicate significance for the SAMP6 mice group.

**Figure 4 pharmaceuticals-16-01350-f004:**
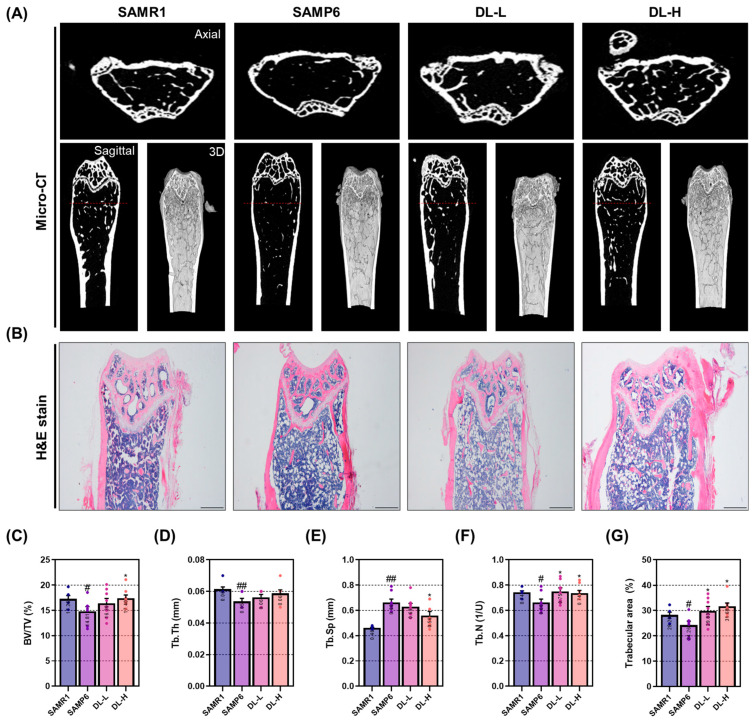
DL prevents loss of bone density and deterioration of bone microarchitecture in senile osteoporosis mice. (**A**) Micro-CT images were captured to analyze alterations in bone mineral density and microstructure of the femur. (**B**) Furthermore, quantitative changes in cancellous bone were observed through H&E staining after paraffin embedding. Micro-CT software (version 1.6.10.1) was utilized to analyze bone microstructural factors, including (**C**) BV/TV, (**D**) Tb.Th, (**E**) Tb.Sp, and (**F**) Tb.N. (**G**) ImageJ software 1.46 was employed to measure the area of cancellous bone in the stained tissue. Each mouse group consisted of 8 mice. Dots in each graph represent individual experimental data values. Data are presented as mean ± SEM. ## *p* < 0.01 and # *p* < 0.05 indicate significance for the SAMR1 mice group; * *p* < 0.05 indicates significance for the SAMP6 mice group.

**Figure 5 pharmaceuticals-16-01350-f005:**
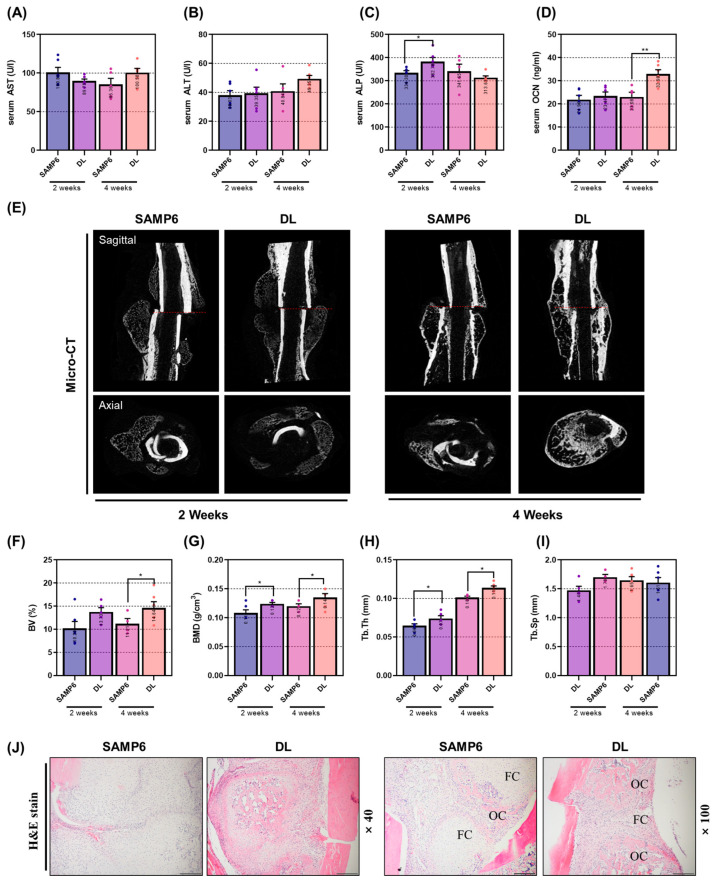
DL promotes bone union through callus formation and positive changes in serum bone morphogenic factors. (**A**,**B**) Serum levels of AST/ALT, known as indicators of hepatotoxicity, were measured using ELISA. ELISA was also employed to measure serum levels of bone-related factors such as (**C**) ALP and (**D**) OCN. (**E**) Micro-CT images were captured to analyze alterations in bone union of the femur. Micro-CT software was utilized to analyze microstructural factors, including (**F**) BV, (**G**) BMD, (**H**) Tb.Th, and (**I**) Tb.Sp. (**J**) Quantitative changes in callus were observed through H&E staining after paraffin embedding. Each mouse group consisted of 8 mice. Dots in each graph represent individual experimental data values. Data are presented as mean ± SEM. ** *p* < 0.01 and * *p* < 0.05 indicates significance for each SAMP6 group.

**Figure 6 pharmaceuticals-16-01350-f006:**
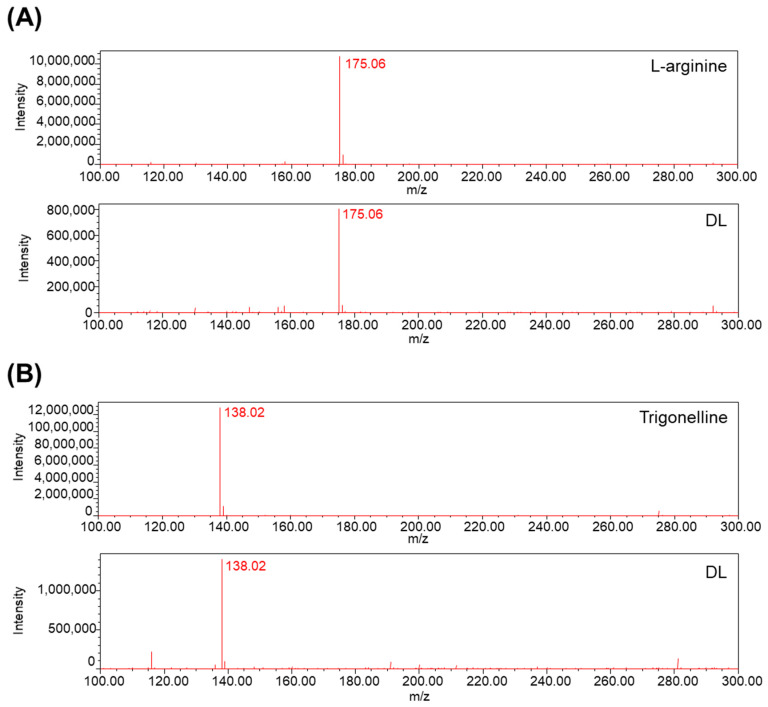
Analysis peaks of DL and indicator components were detected through LC-MS. (**A**) L-arginine and L-arginine in DL and (**B**) Trigonelline and Trigonelline contained in DL were analyzed in the mass range *m*/*z* 100–300.

**Table 1 pharmaceuticals-16-01350-t001:** Information of primary antibody used for Western blot.

Source	Marker Name	Cat No.	Dilution Ratio	Protein Size (kDa)
Mouse	Anti-BMP-2	ab14933	1:1000	45
	Anti-Osterix	ab209484	1:2000	45
	Anti-p-Smad1/5	9516S	1:1000	60
	Anti-Smad1/5/9	ab80255	1:1000	52
	Anti-RUNX2	ab76956	1:1000	57
	Anti-Actin	sc-8432	1:1000	42

**Table 2 pharmaceuticals-16-01350-t002:** Primer sequences used for RT-PCR.

Source	Gene Name	Sequence (5′-3′)	Accession No.	Tm (°C)	Base Pair
Mouse	*Ctnnb1* (β-catenin)	F: TGC TGA AGG TGC TGT CTG TC R: CTG CTT AGT CGC TGC ATC TG	NM_001165902.1	59	158
	*Wnt10b* (Wnt10b)	F: TTC TCT CGG GAT TTC TTG GAT TC R: TGC ACT TCC GCT TCA GGT TTT C	NM_011718.2	59	118
	*Dvl2* (DVL2)	F: GCT TCC ACA TGG CCA TGG GC R: TGG CAC TGC TGG TGA GAG TCA CAG	[35]	64	227
	*Ccnd1* (Ccnd1)	F: GAA GGA GAT TGT GCC ATC R: TTC TTC AAG GGC TCC AGG	[35]	55	346
	*Gapdhs* (GAPDH)	F: ACT TTG TCA AGC TCA TTT CC R: TGC AGC GAA CTT TAT TGA TG	NM_008084.3	58	267
	*Alpl* (ALP)	F: CGG GAC TGG TAC TCG GAT AA R: TGA GAT CCA GGC CAT CTA GC	NM_001287172.1	55	208
	*Bglap* (OCN)	F: GCA ATA AGG TAG TGA ACA GAC TCC R: GTT TGT AGG CGG TCT TCA AGC	NM_001032298.3	59	147
	*Ibsp* (BSP)	F: AAA GTG AAG GAA AGC GAC GA R: GTT CCT TCT GCA CCT GCT TC	NM_008318.3	53	215
	*Col1a1* (COL1)	F: GCT CCT CTT AGG GGC CAC T R: CCA CGT CTC ACC ATT GGG G	NM_007742.4	60	103

## Data Availability

Data is contained within the article.

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
