# Peer review of "Dolichos Lablab Linné Inhibits Bone Density Loss and Promotes Bone Union in Senile Osteoporosis through Osteogenesis"

_pharmaceuticals, 2023, doi:10.3390/ph16101350_

Round 1

Reviewer 1 Report

The manuscript presents an insightful exploration of DL's pharmacological effects on senile bone disease. The comprehensive analysis covers multiple aspects of DL's impact, including osteoblast differentiation, calcified nodule formation, bone density preservation, and bone union enhancement. The article, however, would benefit from a clearer organization and concise language. The current arrangement of figures presents some issues, and there's a need for better organization in the literature where the numbering appears to be a bit confusing (Discusion's section). Specifically, it should be noted that there is an issue with Figure 5. In addition, certain sections could be further refined to enhance clarity and coherence. The authors must carefully revise the current version and organise their material in a more straightforward way. Please see the attached .pdf file which is accompanied by several notes.

English is more or less fine

Author Response

We would like to thank reviewers for their positive and very helpful comments to improve our manuscript. In response to the reviewer, we have addressed the comments as follow:

Reviewer 1.

The manuscript presents an insightful exploration of DL's pharmacological effects on senile bone disease. The comprehensive analysis covers multiple aspects of DL's impact, including osteoblast differentiation, calcified nodule formation, bone density preservation, and bone union enhancement. The article, however, would benefit from a clearer organization and concise language.

The current arrangement of figures presents some issues, and there's a need for better organization in the literature where the numbering appears to be a bit confusing (Discusion's section).

  • We reviewed and revised the number of the figure as a whole.
  • Because we include a brief description of the function of the marker in the results part, there are some cases where the numbering in the discussion part is preceded rather than sequentially. However, I kept the number as is because I thought that adding additional references with similar content would mean adding more information that is not necessary for the manuscript. We ask for your understanding.

Specifically, it should be noted that there is an issue with Figure 5. In addition, certain sections could be further refined to enhance clarity and coherence.

  • We confirmed that there was a problem with the notation in Figure 5 and corrected it.

The authors must carefully revise the current version and organise their material in a more straightforward way. Please see the attached .pdf file which is accompanied by several notes.

  • We have reflected the corrections in the PDF you sent us.

Thank you for your comment!

Reviewer 2 Report

The paper analyzes the  anti-osteoporotic effects of Dolichos lalab Linné. The authors highlighted that  Dolichos lalab Linné promoted the formation of osteoblasts and calcified nodules. The paper is interesting, however I have the following comments for the authors: 

-        In line 11 and line 36 the authors state that the treatment for osteoporosis is limited to postmenopausal osteoporosis. The statement is false (as clearly reported in reference 3). 

-        Please rephrase the sentence in lines  39-41

-        Lines 84-86. Figure 1D report only quantification of Alizarin red S dye. Moreover, figure 1D report different DL concentrations from figure 1C, please clarify. Finally, in Figure 1D there are two # over the second histogram.

-        Please rephrase the sentence in lines 139-141.

-        Please correct the “##” in figures 3A, 3F, 3G  3I, 4D, 4E.

-        Line 247 please clarify “collagen etc.”

-     Lines 311-314. It is not clear how these two sentences relate to the previous ones.

Author Response

Responses to Reviewers:

We would like to thank reviewers for their positive and very helpful comments to improve our manuscript. In response to the reviewer, we have addressed the comments as follow:

Reviewer 2.

The paper analyzes the  anti-osteoporotic effects of Dolichos lalab Linné. The authors highlighted that  Dolichos lalab Linné promoted the formation of osteoblasts and calcified nodules. The paper is interesting, however I have the following comments for the authors: 

-        In line 11 and line 36 the authors state that the treatment for osteoporosis is limited to postmenopausal osteoporosis. The statement is false (as clearly reported in reference 3). 

-> Research on osteoporosis in men (osteoblast-related) is perceived as being relatively less important than osteoporosis in postmenopausal women, and I wanted to mention the sluggish research progress in developing treatments. We recognized that our description may have caused misunderstanding, so we modified the sentence. Additionally, we added another reference along with the previous references no.3.

-        Please rephrase the sentence in lines  39-41

-> We modified that sentence. Please review.

-        Lines 84-86. Figure 1D report only quantification of Alizarin red S dye. Moreover, figure 1D report different DL concentrations from figure 1C, please clarify. Finally, in Figure 1D there are two # over the second histogram.

-> We have removed content from the text (regarding Von kossa positive area). The only thing that was quantitatively analyzed was the area of Alizarin red s. In Figure 1D, ## (P<0.01) is correct, and the meaning of that notation has been added to the figure legend.

-        Please rephrase the sentence in lines 139-141.

-> We reviewed the content and recognized that the process of notating abbreviations could cause confusion to readers. Accordingly, the notation of abbreviations related to SOST and DKK-1 was moved to the first place, and the sentences could be checked intuitively.

-        Please correct the “##” in figures 3A, 3F, 3G  3I, 4D, 4E.

-> We indicated the relevant information in the figure legend.

-        Line 247 please clarify “collagen etc.”

-> We have corrected that sentence.

-   Lines 311-314. It is not clear how these two sentences relate to the previous ones.

-> That sentence appears to be correct at the beginning of the following paragraph. The preceding sentences and paragraphs were divided and combined with the following paragraphs. Through this, information related to bone metabolism and aging was grouped into one paragraph.

Thank you for your comment!

Reviewer 3 Report

The present study titled "Dolichos lablab Linné inhibits bone density loss and promotes bone union in senile osteoporosis through osteogenesis"  evaluate the efficacy of Dolichos lablab Linné to counteract osteoporosis using mouse and cell lines focusing on osteogenesis. The researchers has carried out a large number of tests and analyzed a large number of markers that allow us to know the processes studied in great detail, although it would be interesting to review the size of the results shown because sometimes they are not well appreciated.

However, there are things that need to be improved. First, the studies use different doses that do not always have the same effect on the markers analyzed. Therefore, the results should be clarified by indicating when dose-response relationships are linear or respond to saturation curves and the ranges within which they occur. In turn, the differences found between markers in this regard should be discussed in the discussion section looking for a possible explanation and, more importantly, indicating which effect may be more relevant taking into account the differences between markers.

On the other hand applicability of the product could be compromised if there is an imbalance between bone resorption and bone formation, but the first process has hardly been evaluated. Although the authors indicate that the product acts by promoting osteogenesis, they should point out this weakness and be cautious when establishing its possible use. However, some micro-CT data can give an idea of the balance between the two processes and therefore could give more value in the discussion taking this into account.

Author Response

Responses to Reviewers:

We would like to thank reviewers for their positive and very helpful comments to improve our manuscript. In response to the reviewer, we have addressed the comments as follow:

Reviewer 3.

The present study titled "Dolichos lablab Linné inhibits bone density loss and promotes bone union in senile osteoporosis through osteogenesis"  evaluate the efficacy of Dolichos lablab Linné to counteract osteoporosis using mouse and cell lines focusing on osteogenesis. The researchers has carried out a large number of tests and analyzed a large number of markers that allow us to know the processes studied in great detail, although it would be interesting to review the size of the results shown because sometimes they are not well appreciated.

However, there are things that need to be improved.

First, the studies use different doses that do not always have the same effect on the markers analyzed. Therefore, the results should be clarified by indicating when dose-response relationships are linear or respond to saturation curves and the ranges within which they occur. In turn, the differences found between markers in this regard should be discussed in the discussion section looking for a possible explanation and, more importantly, indicating which effect may be more relevant taking into account the differences between markers.

  • First of all, we confirmed that there was a problem with the recording of the dose in the cell experiment. For example, the capacity of the western blot band is 12.5~100 ug/ml, but the quantified value is indicated as 25~200 ug/ml. All cell experiments were conducted at 12.5 to 100 ug/ml, and all incorrect information was corrected (Figure 1 and Figure 2).
  • Treatment of DL upregulated all tested indicators (differences in significance existed). Therefore, it was not possible to compare the appropriateness of DL through comparison of pharmacological effects between specific mechanisms or specific markers. We apologize for not reflecting the reviewer's opinion.However, we believed that there were areas that needed to be reviewed between cytotoxicity and pharmacological effects in the overall experiment structure, so we added information on areas of concern and discussed future directions. What has been added is as follows:
  • Collectively, the cell experiment findings indicate that DL may have potential applications in elderly osteoporosis patients and individuals with fractures requiring enhanced osteo-blast activity. DL elicits the expression of calcific nodules and osteogenic factors at a rela-tively low concentration (12.5 µg/ml) and exhibits an upward trend across all concentrations tested. Notably, at 100 µg/ml, a significant impact was observed in all experiments when compared to cells treated with the osteogenic medium. However, it is noteworthy that the cell viability test results indicated an effect on cell viability at 200 µg/ml. This concern is believed to be addressable through subsequent experiments and future in vivo toxicity assessments.

On the other hand applicability of the product could be compromised if there is an imbalance between bone resorption and bone formation, but the first process has hardly been evaluated. Although the authors indicate that the product acts by promoting osteogenesis, they should point out this weakness and be cautious when establishing its possible use. However, some micro-CT data can give an idea of the balance between the two processes and therefore could give more value in the discussion taking this into account.

  • When conducting this study, we also investigated the inhibitory effect of DL on osteoclast differentiation. We will send you the results of the experiment together. The experimental method is briefly described at the last page.TRAP is an osteoclast-specific marker and is a verification method used to verify the effectiveness of inhibition. Experimental results showed that DL did not have a significant effect on osteoclast differentiation and activity.Since the main cause of senile osteoporosis is decreased osteoblast activity, we only intend to describe in the manuscript the effectiveness of DL on osteoblast differentiation. Additionally, because the SAMP6 osteoporosis model is generated by differentiation of MSCs into mast cells rather than osteoblasts, the main research purpose is to verify the effectiveness of osteoblast differentiation. Accordingly, no osteoclast content was added.

  • Methods (TRAP assay): To determine the anti-osteoclastogenesis effect of DL, 5,000 RAW 264.7 cells were seeded in 96-well-plates and stabilized for 24 h. RANKL (100 ng/ml) and various concentrations of DL (12.5, 25, 50 and 100 μg/ml) were treated and reacted for 5 days. The culture medium was exchanged on days 2 and 4 with the same medium. The cells were washed with DPBS and fixed with 4% formalin at room temperature for 1 h, and TRAP staining was performed using a TRAP staining kit at 37°C for 1 h. Cells containing three or more nuclei were defined as TRAP-positive cells, and TRAP staining was performed using an inverted light microscope (Olympus Corporation; magnification, ´100). To measure the activity of TRAP in the culture medium, the same amount of TRAP solution (4.93 mg 4-nitrophenyl phosphate disodium salt hexahydrate (PNPP) in 850 μl of 0.5 M acetate solution and 150 μl of tartrate solution) was mixed with the culture medium on the 5th day and reacted for 1 hour. Thereafter, the reaction was intermediated with 0.5M NaOH, and the absorbance was measured at 405 nm using an ELISA reader.

Thank you for your comment!
